# Factors Affecting CPAP Adherence in an OSA Population during the First Two Years of the COVID-19 Pandemic

**DOI:** 10.3390/healthcare12171772

**Published:** 2024-09-05

**Authors:** Dimosthenis Lykouras, Eirini Zarkadi, Electra Koulousousa, Olga Lagiou, Dimitrios Komninos, Argyris Tzouvelekis, Kyriakos Karkoulias

**Affiliations:** Department of Respiratory Medicine, University Hospital of Patras, 26500 Patras, Greece; lykouras@upatras.gr (D.L.); edzarka@gmail.com (E.Z.); komninos312@gmail.com (D.K.); argyris.tzouvelekis@gmail.com (A.T.)

**Keywords:** CPAP, OSA, CPAP adherence, telemedicine

## Abstract

**Background:** Obstructive sleep apnea (OSA) is a common disorder associated with major cardiovascular and neurocognitive sequelae. Continuous positive airway pressure (CPAP) is the standard treatment for OSA. The aim of this study was to investigate the prevalence and associations of long-term CPAP adherence in newly diagnosed OSA patients. **Methods:** We enrolled patients who were diagnosed with OSA during the COVID-19 pandemic. Adherence was defined as CPAP use ≥4 h per night on ≥70% of nights over 30 consecutive days. Patient demographics were retrieved from medical records, and CPAP adherence at 6 months and 1 year after initiation was monitored. **Results:** Overall, 107 patients were included in the analysis. A number of 73 (68%) and 63 (59%) patients were adherent to CPAP treatment at 6 months and 12 months accordingly. Among the factors examined and analyzed (age, gender, BMI, Apnea–Hypopnea Index (AHI)), no significant correlation was found. Further analysis revealed the potential role of comorbidities. CPAP compliance at 6 months was shown to be associated with better CPAP adherence at 12 months. **Conclusions:** CPAP adherence at 6 months is correlated to long-term adherence to treatment. Therefore, early close follow-up is important. Further prospective studies are needed to identify other potential predictors.

## 1. Introduction

Modern society and technological advances render our lifestyle incompatible with good sleep [1]. Obstructive sleep apnea (OSA) is a common sleep disorder that affects up to 23% of middle-aged women and 50% of middle-aged men [2]. It is characterized by repeated collapses of the upper airway, causing interrupted sleep, lower oxygen levels, and increased sympathetic activity, leading to high blood pressure. OSA may also result in cognitive issues and daytime sleepiness, which can significantly affect quality of life [3].

Continuous positive airway pressure (CPAP) is widely recognized as the primary treatment option for OSA. Despite ongoing debates about its impact on cardiovascular outcomes, CPAP is proven to reduce daytime sleepiness and enhance mood and quality of life [4]. A patient with OSA may need to use CPAP for at least 4 h a night to experience a reduction in daytime symptoms, especially sleepiness and neurocognitive function, and reduce the risk of developing cardiovascular and metabolic comorbidities. An overall CPAP use of >4 h per night on most nights during a week (>70% of nights) marks optimal CPAP adherence in most studies [5]. However, its efficacy can be hampered by poor adherence to the therapy [6].

Facing difficulties with CPAP therapy is a frequent issue seen in clinical settings. Research indicates that adherence to CPAP therapy usually ranges from 30 to 60%. In cases of mild OSA, where the symptoms are typically less severe and thus easier to manage, adherence to CPAP therapy may potentially be even lower [7].

A myriad of both medical and non-medical factors may contribute to configuring CPAP adherence and numerous studies have attempted to investigate the determinants. Several demographic variables, such as age, gender, and body mass index (BMI), are reported to have an effect on compliance. The severity of OSA, as indicated by a high Apnea–Hypopnea Index (AHI) and excessive daytime sleepiness, has been associated with better adherence to CPAP therapy. The presence of comorbidities, as well as socioeconomic factors, such as economic, educational, and civil status, have also been shown to interfere with treatment compliance. Moreover, device-related factors, which can include physical discomfort while using the device, financial and insurance reimbursement issues, and the associated psychological stress [8], have also been highlighted. Finally, early compliance is pointed out as a positive predictive factor, while interventions such as the use of telemedicine may favorably influence long-term adherence [9].

Therefore, a comprehensive and individualized approach is required to optimize patient care and mitigate the risk of treatment failure [10]. However, even though the issue has been extensively addressed, several studies have yielded conflicting results, and there is still no consensus on which specific factors are associated with CPAP adherence.

Adding to the above, since the onset of the COVID-19 pandemic in 2020, the world has rapidly transformed. This transformation has particularly affected patients with sleep-disordered breathing conditions, especially OSA, who face potentially more severe COVID-19 outcomes due to the shared risk factors and comorbidities of both diseases. Initially, continuous positive airway pressure (CPAP) therapy, the primary option for moderate to severe OSA, was marked as a high-risk aerosol-generating procedure causing fear of potential COVID-19 transmission. To address these issues, several scientific societies have issued recommendations on screening, diagnosing, and treating sleep-disordered breathing during the pandemic. The pandemic caused a decrease in patient visits in clinics across Europe, Greece included. Most centers have had to restrict themselves to phone-based follow-ups and handling high-priority cases [11].

The aim of this study was to investigate the prevalence of long-term CPAP adherence in newly diagnosed OSA patients and identify specific factors that are linked to improved adherence. The study time frame was during the first 2 years of the COVID-19 pandemic.

## 2. Materials and Methods

### 2.1. Subjects

This is a retrospective observational study of patients referred to the sleep laboratory of the Department of Respiratory Medicine, University Hospital of Patras, Greece. All patients recruited for the study underwent night polysomnography (NPSG) between January 2020 and December 2022, during the initial phase of the COVID-19 pandemic. Only adult patients who were diagnosed for the first time with OSA and started CPAP treatment were included in the study analysis. Old CPAP users and patients who did not start CPAP treatment for any reason (declined, reimbursement issues, patient preference) were excluded from the analysis. Patients with overt cardiovascular disease and patients with a diagnosis of active cancer at the time of data collection after the phone contact were not included.

The diagnosis of OSA is made when a patient undergoing a diagnostic sleep study has an AHI of 5 or more. Patients were classified by AHI into mild (AHI ≥ 5 and <15), moderate (≥15 and <30), and severe (AHI ≥ 30) OSA. CPAP adherence was defined as at least 4 h/night use on 70% of nights. Patients who had a diagnosis other than OSA, such as insomnia and obesity hypoventilation syndrome, were excluded from the study. We also excluded patients with increased AHI who were prescribed non-invasive ventilation (NIV) for other indications.

It should be noted that the recruitment period was during the initial phase of the COVID-19 pandemic; a time during which mandatory sleep laboratory closures led to limited access for patients to some of our services. Patients initially undergo a sleep study in the laboratory, and then a titration study follows. After CPAP initiation at home, they return to an outpatient clinic, or they receive a follow-up phone call at 1 month to check initial compliance and treatment efficacy. Afterwards, regular visits happen every 6 months, when a prescription for reimbursement is needed as well. Restrictive measures during the pandemic reduced subsequently in-person visits.

The medical records of included patients were accessed, and data regarding demographics (age, gender, profession, marital status, body mass index (BMI)), sleep questionnaires (Epworth Sleepiness Score (ESS), Athens Insomnia Scale (AIS)), and sleep study parameters (Apnea–Hypopnea Index (AHI)) were collected. We did not explore other sleep study parameters, such as the oxygen desaturation index (ODI) or the average desaturation percentage. Sleep studies were scored according to current American Academy of Sleep Medicine (AASM) diagnostic criteria [12]. We also collected medical information regarding chronic concomitant diseases, including diabetes, hypertension and heart failure, chronic obstructive pulmonary disease, and depression. Patients should have been on treatment for at least 3 months to be regarded as suffering from the respective diseases. The CPAP start date was noted as the initial checkpoint for the study.

### 2.2. Data Collection

CPAP usage information was collected through the AirView platform for patients using ResMed CPAP machines. More specifically, patients were checked for adherence to their treatment at 6 months and 12 months after CPAP initiation. Additional information was obtained by a phone call. Patients who were using other CPAP machines were only interviewed by phone call.

The investigators managed to identify CPAP patients and then gathered information regarding CPAP usage and adherence to treatment according to the study protocol from existing information. Afterwards, study participants were contacted via telephone, and structured interviews were performed containing questions on side effects, co-morbidities, lifestyle, and other factors that could possibly have impaired adherence. It should be noted that telephone interviews may be linked to patient reporting bias. We combined these data with information from electronic patient records and other medical records available at our site.

Adherence was defined as adequate use of CPAP for ≥4 h on at least >70% of nights during the last month prior to study checkpoints (before 6 months and before 12 months). Patients were grouped accordingly to adherent and non-adherent.

### 2.3. Statistical Analysis

Statistical analysis was performed using IBM SPSS version 28 (IBM, Armonk, NY, USA), and *p* < 0.05 was statistically significant. All results are presented as mean values and standard deviations (SD) unless stated otherwise. Comparisons between the two groups of patients (compliant at 1 year and non-compliant at 1 year) were made using chi-square for binary variables, *t*-test for normally distributed variables, and Wilcoxon rank test for not normally distributed variables. We performed bivariate analysis to investigate the potential correlations between participants’ characteristics and their adherence to CPAP treatment at 6 months and 12 months after starting their CPAP treatment.

### 2.4. Ethical Considerations

The study has been approved by the Institutional ethical committee and review board (approval code 143/2024). The study was performed according to the Declaration of Helsinki. All participants had provided verbal consent to participate in the study via telephone call or had already provided consent for the online data management platform. The verbal consent was documented by the interviewing physician.

## 3. Results

The overall number of patients who underwent a night study during the studied period was 240. A total of 134 patients were eligible for entry in the study, as they were diagnosed with OSA and started CPAP during the years 2020–2022. As this is a retrospective analysis, the dataset was reduced to 107 patients on CPAP because of corrupt contact information (21 subjects had either given a number that was not receiving or patients were not responding) and unwillingness to participate in the study (6 subjects).

Baseline characteristics of the OSA patients started on CPAP are shown in Table 1. They were divided into two groups according to their adherence to CPAP treatment 1 year after CPAP titration and CPAP initiation. The mean age for the non-adherent and adherent group at 1 year were 53.4 (10.2) and 56.2 (11.0) years, respectively. Both genders were represented, but the number of men was slightly higher, without any impact on data interpretation. The mean BMI (33.7 versus 36.7), AHI (46.8 versus 51.0), and ESS (11.4 versus 10.3) were not significantly different between the groups. Thus, the groups were matched, and any outcome variation was not due to baseline predisposition. A key point in our analysis was the fact that patients both near the site and far away showed similar adherence to CPAP treatment (20 patients near the site city and 23 patients in remote areas in the non-adherent group) (34 near the site city and 30 in the remote area adherent group).

A total number of 73 patients were using CPAP treatment at 6 months after treatment initiation, with an adherence level of 68.2%. The percentage of adherence fell 1 year after treatment initiation and was as high as 59.8%, as demonstrated in Table 1.

We attempted to analyze factors that may result in this important reduction in treatment compliance. Age, gender, weight, height, BMI, Apnea–Hypopnea Index (AHI) at diagnosis, and residence of patients (in the city of the site or in remote areas) were tested in the analysis. None of these factors were shown to be associated with worse adherence to CPAP treatment.

A percentage of 23% of those who discontinued CPAP usage reported that difficulties with its use (noise, mask interface issues, leakages) had urged them to do so. Another 10% of non-compliant patients had stopped their treatment due to financial issues and lack of insurance, thus reflecting an important aspect to be considered, especially in long-term treatments.

We also attempted to explore the role that comorbidities may play in adherence to treatment. Therefore, we tested the effect of diabetes, hypertension, heart failure, COPD, and depression on CPAP adherence at the end of the first year of treatment. However, none of these variables demonstrated statistical significance.

We performed bivariate analysis to further investigate the potential correlations between the patients’ characteristics and their adherence to CPAP treatment at 6 months and 1 year after treatment initiation (Table 2). The analysis revealed some statistically significant correlations that could enable the prediction of CPAP adherence based on the baseline characteristics in our study population. Older age was a predictor in both groups (*p* = 0.049 at 6 months and *p* = 0.029 at 1 year), and increased BMI was a determinant of better adherence at 1 year of CPAP use (*p* = 0.020). As for the comorbidities, a significant correlation was found for CPAP users with depression (*p* = 0.037), who were more likely to have better compliance at 1 year of treatment.

Further multivariate analysis of our data was performed to elucidate the most important factors affecting CPAP adherence at 1 year after treatment initiation. The results showed that patients with optimal adherence at 6 months had a higher likelihood of continuing their treatment (*p* < 0.001). Patients on treatment for depression (*p* = 0.033) were showing better compliance, and patients using telemedicine tools for device monitoring had a tendency (*p* = 0.072) to have improved compliance with their CPAP treatment.

Our study shows that patients showing good compliance at 6 months may also have good CPAP compliance at 1 year (Table 2). Online management tools were not directly associated with improvements in compliance, yet a trend towards improvement was recorded in our data, as patients who did not use available telemedicine tools tended to be non-adherent 1 year after CPAP treatment initiation.

## 4. Discussion

CPAP is the treatment of choice for OSA and is seen as the benchmark. Research shows that CPAP treatment can reduce the risk of cardiovascular disease in patients with moderate to severe OSA, improving their quality of life and daytime sleepiness. CPAP adherence is important for treatment outcomes, but adherence rates have been reported to vary between 30 and 60% in different previous studies.

In our study, we found that 68% of the patients were using CPAP 6 months after treatment initiation. This finding is in line with the 6-month adherence rate of 63.9% and 70.3% reported in previous studies in the field [13,14]. Those patients were using their treatment according to guidelines for CPAP use, that is, at least 4 h every night and at least >70% of nights every month.

A recent study investigating long-term adherence to CPAP treatment in mild OSA showed that only 25.7% of patients continued using their machine 1 year after starting it. Further analysis showed that older patients with lower BMI and the presence of a bed partner had a better chance of remaining on the treatment [15]. A large study, in a population in Asia that recruited OSA patients with mild up to severe disease, revealed that a total of 78.5% of CPAP users remained adhering to their treatment 1 year after treatment had started. They also concluded that trial treatment of 1 month was improving overall treatment adherence [16]. Our results from our single-site population revealed quite a high percentage of adherence to CPAP treatment in the patients, which is comparable to studies recruiting patients at all stages of disease severity.

CPAP therapy can be intrusive for patients, and low adherence can be a major problem. One of the first studies in the field of CPAP adherence was published back in 1993 [17]. CPAP was available in clinical practice for less than 10 years at that time. In this study, memory chips were used to accurately track adherence data. They analyzed key metrics, including mean nightly usage time, the number of nights the therapy was used, and side effects. Their findings suggested that using the therapy for at least 4 h on 70% of nights was a reliable indicator of treatment compliance. In another study investigating CPAP adherence patterns, the authors discovered two groups of CPAP users among their participants. The “intermittent” users tended to skip 1–7 nights per week and used the therapy for fewer hours each night. On the other hand, the “consistent” users applied their devices on 90% of nights or more and typically used them for longer periods [18].

Our study shows that adherence to CPAP at 6 months after treatment initiation is predictive of CPAP adherence at 1 year. This finding is in accordance with previous studies showing that if CPAP adherence is achieved early, CPAP compliance can last [19]. In the SAVE (Sleep Apnea Cardiovascular Endpoints) study, only compliance at 1 month and the presence of side effects of CPAP therapy were independent predictors of its use at 12 months. Similarly, previous studies have reported that CPAP usage patterns within the first week are predictive of long-term use [20].

The examination of other factors that could be correlated to optimal CPAP adherence at 1 year, such as sex, age, increased BMI or worse AHI, did not reveal any significant results. However, a further bivariate logistic regression analysis showed that CPAP users who were older and had a higher BMI were more likely to comply with treatment at 1 year. A retrospective observational study of 1339 OSA patients found no association between CPAP adherence and demographic or polysomnographic variables, which is in accordance with our negative results. The association between AHI and CPAP adherence was not confirmed in a study of 188 patients with moderate to severe OSA [21].

Hypersomnolence, as indicated by high ESS, has been proposed as a predictor of compliance. In our analysis, we found no significant correlation. In agreement with our results, a previous study also predicted no association with ESS, which can also be justified by the fact reported by some authors that ESS score is poorly correlated with OSA severity [22].

Moreover, several studies have suggested that rural residence might be a potential barrier for OSA patients to diagnosis and care. An important prospective cohort study of 242 uncomplicated patients with OSA, urban versus rural residence, was not associated with CPAP use or adherence [23]. Correspondingly, we found no significant difference between patients coming from the city and those from remote areas.

Another issue to be addressed is comorbidities in patients under CPAP treatment for OSA. Diseases, including diabetes, hypertension, COPD and depression, are highly prevalent in the OSA population, and we intended to determine their role in adherence to CPAP treatment. Interestingly, patients under treatment for depression were more likely to be compliant with CPAP treatment one year after starting it. However, a large-scale study of a similar population in Greece did not demonstrate a potential role of any comorbidity in adherence to treatment.

An important finding of our study is that patients who demonstrated good compliance at 6 months may also have good CPAP compliance at 1 year. Therefore, close monitoring during the first months of CPAP usage is of paramount importance for long-term compliance. Future studies may consider shorter monitoring periods of 3 months in the initial phases of CPAP treatment to further clarify this issue.

The online management tool that was used by some of our patients, ResMed Airview, was also not found to improve compliance in our analysis. Indeed, previous studies have reported comparable findings. A multinational study indicated that telemonitoring did not improve 1-month and 3-month CPAP adherence compared with usual care, while in a cohort of 120 patients with OSA, the usage of telemedicine during the critical habituation phase for CPAP did not alter daily CPAP use or treatment adherence at 3 and 12 months [24]. In contrast, a recent meta-analysis of 11 studies with 1358 revealed that telemedicine interventions may improve CPAP adherence in patients with OSA compared to no intervention [25]. However, although the efficacy of telemonitoring systems has been inconsistent, most trials have shown that such tools could prove more cost-effective and source-saving. Meanwhile, especially in a setting simulating the COVID-19 pandemic, with reduced footfall in healthcare facilities, telemonitoring could prove of significant value. Moreover, in our region in Greece, we treat patients from remote areas of the islands, and telemedicine is of great value in this setting as well.

A large prospective study from Greece that investigated the impact of the COVID-19 pandemic on the CPAP adherence of patients already on CPAP concluded that it did not actually affect device usage. Moreover, the use of telemedicine was important to overcome barriers caused by lockdowns in order to facilitate regular follow-ups and increase CPAP treatment adherence [11].

The treatment rejection rate at 1 year in our study is acceptable. The reasons for treatment discontinuation included difficulties with CPAP daily use, such as noise, mask interface issues, and leakages. Another important aspect of CPAP discontinuation is the relatively high cost of obtaining a CPAP machine either privately or through insurance coverage. A substantial minority (10%) of patients non-compliant with CPAP prematurely discontinued treatment due to financial issues and lack of insurance.

Our study was designed to demonstrate CPAP adherence among patients during the years of the pandemic. At that time, several lockdowns in Greece appreciably affected sleep laboratory services and sleep clinics. Nevertheless, regular follow-up visits and phone consultations may have resulted in quite high CPAP adherence in our cohort. Furthermore, several studies have reported ameliorated CPAP adherence during the COVID-19 pandemic. In a cohort of 7485 patients with OSA, the investigators reported a 3.9% increase in adherence from a mean value of 386 min per night pre-COVID-19 to 401 min per night during lockdown [26]. Another study also observed that improved CPAP adherence in severe OSA patients during the COVID-19 lockdown was more pronounced in women and younger and pre-lockdown CPAP adherers [27].

Our study has some limitations. Firstly, this is a retrospective study; therefore, missing information may have reduced our initial sample size. Secondly, phone interviews may have intrinsic defects with regard to the accuracy of information derived from patients. Thirdly, the retrospective design did not allow us to have correlations with other comorbidities that could affect adherence levels, such as hypertension and diabetes. The last remark is that our study is a single-site retrospective study that may have selection bias caused by the availability of some technologies or geographic distribution.

## 5. Conclusions

Effective CPAP treatment can be affected by several factors. Patients who were adherent to CPAP treatment at 6 months were more likely to continue using their device and on a long-term basis, thus, showing sustained adherence at 1 year and subsequently a lower drop-out rate. Other factors, including OSA severity, BMI, or sleepiness severity, were not significantly correlated with long-term CPAP adherence. Further prospective studies may be needed to clarify other potential predictors of treatment compliance.

## Figures and Tables

**Table 1 healthcare-12-01772-t001:** Baseline characteristics of study subjects.

	CPAP Non-Adherent at 1 Year (n = 43)	CPAP Adherent at 1 Year (n = 64)	All Patients (n = 107)	*p*-Value
Age (years)	53.4 (10.2)	56.2 (11.0)	55.2 (10.7)	0.434
Gender (M/F)	35/8 (81.3%/18.7%)	51/13 (79.6%/20.4%)	86/21 (80.3%/19.7%)	0.516
Height (cm)	174.3 (8.1)	173.6 (10.0)	173.9 (9.2)	0.270
Weight (kg)	102.1 (19.3)	109.8 (21.0)	106.7 (20.6)	0.988
BMI (kg/m^2^)	33.7 (7.0)	36.7 (8.1)	35.5 (7.8)	0.238
AHI (events/h)	46.8 (24.4)	51.0 (27.4)	49.3 (26.2)	0.215
Epworth Sleepiness Scale (ESS) score	11.4 (4.9)	10.3 (5.1)	10.7 (5.0)	0.920
Athens Insomnia Scale (AIS) score	7.8 (4.4)	7.2 (3.9)	7.4 (4.1)	0.406
Residence (site city/remote)	20/23 (46.5%/53.5%)	34/30 (53.1%/46.9%)	54/53 (50.4%/50.6%)	0.318
Diabetes	7 (16.2%)	15 (23.4%)	22 (20.5%)	0.467
Hypertension	24 (55.8%)	30 (46.8%)	54 (50.4%)	0.432
Heart failure	14 (32.5%)	18 (28.1%)	32 (29.9%)	0.670
COPD	9 (20.9%)	15 (23.4%)	24 (22.4%)	0.817
Depression	6 (13.9%)	18 (28.1%)	24 (22.4%)	0.102

Values are presented as mean (SD) unless stated otherwise.

**Table 2 healthcare-12-01772-t002:** CPAP online management and CPAP usage at 6-month correlation to overall CPAP adherence at 1 year.

	CPAP Non-Adherent at 1 Year (n = 43)	CPAP Adherent at 1 Year (n = 64)	*p*-Value
CPAP online management (yes/no)	16/27 (37.2%/62.8%)	30/34 (46.8%/53.2%)	0.215
CPAP adherence at 6 months (yes/no)	11/32 (25.6%/74.6%)	62/2 (96.8%/3.2%)	<0.001

## Data Availability

Data are contained within the article.

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
