# Peer review of "Factors Affecting CPAP Adherence in an OSA Population during the First Two Years of the COVID-19 Pandemic"

_healthcare, 2024, doi:10.3390/healthcare12171772_

Round 1

Reviewer 1 Report (Previous Reviewer 2)

Comments and Suggestions for Authors

After reviewing the new version of the paper, I still have some considerations to make.

The study was carried out during the pandemic, but no factor that directly depends on it is analyzed. There is only a temporal relationship and a period prior to the pandemic is not analyzed or compared, so the factors that affect adherence could be the same. The objective does not establish any aspect regarding Covid-19, only that it was carried out in the first two years of the pandemic. The conclusions do not mention the Covid-19 pandemic, so perhaps the reference to Covid-19 should be removed from the title.

The methodology and results still do not indicate whether all patients who entered the study were included and do not indicate anything about possible losses to follow-up.

The presentation of the tables should be improved. Table 1 should contain one more column, with the overall data of all patients included in the study. The percentages should be given with a decimal, as should the quantitative variables (this aspect should also be corrected in the text). Many of the data that appear between lines 186 and 202 should be removed when presented in the tables.

I do not understand Table 2. If the "p" is presented, the data for the variables should be presented. The authors do a bivariate analysis at 6 and 12 months. The study at twelve months is already presented in Table 1. How is it possible that the significance is completely different in Table 2 and Table 1 for adherence at 12 months? I do not understand it.

If the authors want to present the data at 6 months, they should add two more columns in Table 1 (with adherents and non-adherent at 6 months).

In the results section, only the data that emerge from the study should be included and not the authors' comments on them. These comments should go in the discussion section (e.g. lines 184-186, 208-210).

I find Table 3 confusing; the percentages should be included.

Conclusions should be derived exclusively from the results and should not contain opinions of the authors. These comments should be removed from the conclusions (for this purpose, the discussion section is available).

Author Response

Comment: After reviewing the new version of the paper, I still have some considerations to make.
Reply: Thank you for your re-evaluation. Green high-lighted text is new text.

Comment: The study was carried out during the pandemic, but no factor that directly depends on it is analyzed. There is only a temporal relationship and a period prior to the pandemic is not analyzed or compared, so the factors that affect adherence could be the same. The objective does not establish any aspect regarding Covid-19, only that it was carried out in the first two years of the pandemic. The conclusions do not mention the Covid-19 pandemic, so perhaps the reference to Covid-19 should be removed from the title.
Reply: We have already replied to similar comment in previous version and the title had been changed. As already said, the COVID-19 period affected the number of our patients.

Comment: The methodology and results still do not indicate whether all patients who entered the study were included and do not indicate anything about possible losses to follow-up.
Reply: We added some text in results. The initial number of patients was 240. We have already made clear in our text that patients lost to follow-up were not included in the analysis. The analysis dataset included 107 with full data on all follow-ups (lines 156-159). 

Comment: The presentation of the tables should be improved. Table 1 should contain one more column, with the overall data of all patients included in the study. The percentages should be given with a decimal, as should the quantitative variables (this aspect should also be corrected in the text). Many of the data that appear between lines 186 and 202 should be removed when presented in the tables.
Reply: Thank you for your comment. We have made suggested changes. We have improved the content in our table based on your suggestions.

Comment: I do not understand Table 2. If the "p" is presented, the data for the variables should be presented. The authors do a bivariate analysis at 6 and 12 months. The study at twelve months is already presented in Table 1. How is it possible that the significance is completely different in Table 2 and Table 1 for adherence at 12 months? I do not understand it. If the authors want to present the data at 6 months, they should add two more columns in Table 1 (with adherents and non-adherent at 6 months).
Reply: The p-values in Table 2 come from bivariate analysis of all potential contributing factors. We did not present data at 6 months to reduce complexity of the table.

Comment: In the results section, only the data that emerge from the study should be included and not the authors' comments on them. These comments should go in the discussion section (e.g. lines 184-186, 208-210).
Reply: We have changed the respective text.

Comment: I find Table 3 confusing; the percentages should be included.
Reply: We have made the suggested changes.

Comment: Conclusions should be derived exclusively from the results and should not contain opinions of the authors. These comments should be removed from the conclusions (for this purpose, the discussion section is available).
Reply: We have made the suggested changes.

Reviewer 2 Report (New Reviewer)

Comments and Suggestions for Authors

Thank you for considering me for reviewing this manuscript entitled: "Factors affecting CPAP adherence in an OSA population during the first two years of the COVID-19 pandemic."

Overall, the study is well-written. However, the scientific soundness is fairly low, as it addresses issues that have already been discussed in previous literature.

For example, a recent article by Qiao et al. already identified compliance at the 1st, 3rd, 6th, and 9th months as a significant predictor of long-term (1 year) adherence to CPAP.

Other factors I would suggest to improve:

  • Please disclose other major polysomnographic variables (e.g., ODI, NadirO2, and so on).

  • Line 123: Clearly state that telephonic interviews imply information bias.

  • It appears that depression is a major predictor for long-term compliance (but not short-term). You should better discuss this finding, as it is not supported by current literature.

  • Improve the English language in lines 68-70 and 102-105.

Comments on the Quality of English Language

see comments to Authors

Author Response

Comment: Thank you for considering me for reviewing this manuscript entitled: "Factors affecting CPAP adherence in an OSA population during the first two years of the COVID-19 pandemic."
Reply: Thank you for your time and suggestions. Green high-lighted text is new text.

Comment: Overall, the study is well-written. However, the scientific soundness is fairly low, as it addresses issues that have already been discussed in previous literature. For example, a recent article by Qiao et al. already identified compliance at the 1st, 3rd, 6th, and 9th months as a significant predictor of long-term (1 year) adherence to CPAP.
Reply: We can understand your concern, but we feel that our study supports existing knowledge.

Comment: Other factors I would suggest to improve: Please disclose other major polysomnographic variables (e.g., ODI, NadirO2, and so on).
Reply: We have made the suggested changes.

Comment: Line 123: Clearly state that telephonic interviews imply information bias.
Reply: We have made the suggested changes.

Comment: It appears that depression is a major predictor for long-term compliance (but not short-term). You should better discuss this finding, as it is not supported by current literature.
Reply: Our data has lead to this finding. We suppose that it could be due to the sample size. Larger studies may lead to safer conclusion on this matter.

Comment: Improve the English language in lines 68-70 and 102-105.
Reply: We have made the suggested changes.

Round 2

Reviewer 1 Report (Previous Reviewer 2)

Comments and Suggestions for Authors

I thank the authors for the effort they have made to satisfactorily answer the reviewer's recommendations

The authors have not satisfactorily answered a point raised in the previous review:

 I do not understand Table 2. The authors do a bivariate analysis at 6 and 12 months. The study at twelve months is already presented in Table 1. How is it possible that the significance is completely different in Table 2 and Table 1 for adherence at 12 months? I do not understand it.

Author Response

Comment: I thank the authors for the effort they have made to satisfactorily answer the reviewer's recommendations.
Reply: Thank you for your suggestions.

Comment: The authors have not satisfactorily answered a point raised in the previous review: I do not understand Table 2. The authors do a bivariate analysis at 6 and 12 months. The study at twelve months is already presented in Table 1. How is it possible that the significance is completely different in Table 2 and Table 1 for adherence at 12 months? I do not understand it.
Reply: The dataset in the analysis is the same across the manuscript, and all statistics were done properly using SPSS as already stated in the methods sections. P-values indicate the verification of a hypothesis or tendency depending on the test you run. Therefore, different comparisons may come with different p-values. But, we decided to remove the table to avoid confusion, as per your suggestion.

Reviewer 2 Report (New Reviewer)

Comments and Suggestions for Authors

Author made precise corrections following my comments.

I just have few further concerns

- Table 2: you should perform multivariate analysis for significant variables to exclude possible confounding factors. I also would include "c-PAP adherence ad 6months" in the multivariate - as it is the mayor finding of your study and shouldn't be only supported by a cross tab (risk of bias)

- Lines 109-111: Disclose this issue also in the limitations of the study

Author Response

Comment: Author made precise corrections following my comments.
Reply: Thank you for your suggestions.

Comment: I just have few further concerns. Table 2: you should perform multivariate analysis for significant variables to exclude possible confounding factors. I also would include "c-PAP adherence ad 6months" in the multivariate - as it is the mayor finding of your study and shouldn't be only supported by a cross tab (risk of bias).
Reply: We have now made suggested changes and performed multivariate analysis including CPAP adherence at 6 months. Thank you for your suggestion to improve the manuscript.

Comment: Lines 109-111: Disclose this issue also in the limitations of the study.
Reply: We have made the suggested change.

This manuscript is a resubmission of an earlier submission. The following is a list of the peer review reports and author responses from that submission.

Round 1

Reviewer 1 Report

Comments and Suggestions for Authors

This study was to investigate the prevalence and associations of long-term CPAP adherence in newly diagnosed OSA patients. On the one hand, the results of this study are too few. On the other hand, the phrase "during the COVID-19 pandemic" in the title only refers to the period from 2020 to 2022 and does not compare the CPAP adherence with the COVID-19 pandemic and non pandemic periods. In addition, the innovation of the article is insufficient to attract readers.

Comments on the Quality of English Language

In abstract, 73 (68%) and 63 (59%) patients were 17  adherent to CPAP treatment at 6th month and 12th month accordingly. Individual numbers should not be written at the beginning of a sentence.

Author Response

Comment: This study was to investigate the prevalence and associations of long-term CPAP adherence in newly diagnosed OSA patients. On the one hand, the results of this study are too few.
Reply: We understand your point. The study population refers to the first 2 years of COVID-19 pandemic, when all sleep labs in Greece were facing periodic closures and restrictions betwenn 2020-2022. We included all patients with full records from our laboratory.

Comment: On the other hand, the phrase "during the COVID-19 pandemic" in the title only refers to the period from 2020 to 2022 and does not compare the CPAP adherence with the COVID-19 pandemic and non pandemic periods.
Reply: Indeed. In this study we did not make any correlations with previous years, when our laboratory was performing studies without restrictions.

Comments on the Quality of English Language: In abstract, “73 (68%) and 63 (59%) patients were 17 adherent to CPAP treatmentat 6th month and 12th month accordingly”. Individual numbers should not be written at the beginning of a sentence.
Reply: Thank you for your comment. We have made suggested changes in the respective points in our text.

Reviewer 2 Report

Comments and Suggestions for Authors

I thank the authors for the opportunity they have given me to read this interesting paper.

I have quite a few doubts about the methodology used. I do not understand the influence of the COVID19 pandemic on the results and the methodology used.

Although this study was carried out during the pandemic, the authors should indicate what monitoring protocol they had before the pandemic and during the pandemic. What changed? Did they monitor them more frequently? If so, was it in person or by telephone?

How did you determine the sample size necessary for the study results to be consistent?

Being a retrospective study, did the study begin in 2020 with a specific follow-up protocol for the study or was it the usual protocol to date?

The authors only describe that patients were made 2 calls at 6 and 12 months. Wasn't there any more from anyone? Were all calls made by the investigators and were they provided with the same information? Was this information protocolized?

Since CPAP was introduced, were patients not called before 6 months to find out if there was any adaptation problem, for example with the mask, that could be solved?

For patients who were not compliant at 6 months, were CPAP withdrawn or were they recommended and insisted that they use it? If it was the latter, was it evaluated after a year?

Wasn't the type of mask (nasal, etc.) analyzed if it influenced adhesion?

In the results, I advise removing the numerical data already reflected in the tables from the text. The authors indicate numerical data without decimals, with one decimal or with two decimals. I advise unifying all of them and always indicating with one or two decimal places (including percentages).

The comments that the authors make in the results section about them should be eliminated, such as in line 156 "As expected...", as well as the paragraph that begins in line 169. In the results section only They must provide the data and the authors must not make any comments on them; Comments should be reserved for discussion.

In tables 1 and 2, the percentages of the categorical variables should be entered.

No relationship test is done to show causality or influence, such as providing OR data.

Logically, patients who have been on CPAP for 6 months are more likely to continue with it. I don't understand the phrase: "CPAP compliance at 1 year is higher in patients already fully compliant at 6 months." Higher than what? Patients who were not compliant at 6 months were already off cpap.

Author Response

Comment: I thank the authors for the opportunity they have given me to read this interesting paper.
Reply: Thank you for your comment.

Comment: I have quite a few doubts about the methodology used. I do not understand the influence of the COVID19 pandemic on the results and the methodology used.
Reply: We understand your point. The study population refers to the first 2 years of COVID-19 pandemic, when all sleep labs in Greece were facing periodic closures and restrictions betwenn 2020-2022. We included all patients with full records from our laboratory. The restrictive measures resulted in limited patient numbers.

Comment: Although this study was carried out during the pandemic, the authors shouldindicate what monitoring protocol they had before the pandemic and during thepandemic. What changed? Did they monitor them more frequently? If so, was it inperson or by telephone?
Reply: Thank you for the clarification. We have added some text. Before pandemic restrictions patients using ResMed devices were monitored by Airview and followed-up in outpatient clinics at 1-2 months after CPAP initiation, at 6 months and then every 6 months in outpatient consulation. Phone calls are also used in some patients who cannot attend an outpatient clinic. During the pandemic we continued with our plan, but of course with limited numbers as described above.

Comment: How did you determine the sample size necessary for the study results to be consistent?
Reply: Our study was designed as an observational study and we included all patients with consistent and full records.

Comment: Being a retrospective study, did the study begin in 2020 with a specific follow-uprotocol for the study or was it the usual protocol to date?
Reply: As explained before, patients attend a clinic at 1-2 months after CPAP intitiation and return every 6 months for follow-up and prescription for their device for reimbursement.

Comment: The authors only describe that patients were made 2 calls at 6 and 12 months. Wasn't there any more from anyone? Were all calls made by the investigators andwere they provided with the same information? Was this information protocolized?
Reply: Usual follow-up is done every 6 months.

Comment: Since CPAP was introduced, were patients not called before 6 months to find out if there was any adaptation problem, for example with the mask, that could be solved?
Reply: As explained before, patients attend a clinic at 1-2 months after CPAP intitiation and return every 6 months for follow-up and prescription for their device for reimbursement. Patients that discontinued before 1 month either attend a clinic or never show up.

Comment: For patients who were not compliant at 6 months, were CPAP withdrawn or were they recommended and insisted that they use it? If it was the latter, was it evaluated after a year?
Reply: Patients attend a clinic and have new CPAP prescriptions or they do not show up and are asked about their status. Information are stored.

Comment: Wasn't the type of mask (nasal, etc.) analyzed if it influenced adhesion?
Reply: Most patients use a nasal interface and only a few cases had nasal pillows. We did not analyze this parameter.

Comment: In the results, I advise removing the numerical data already reflected in the tables from the text. The authors indicate numerical data without decimals, with one decimal or with two decimals. I advise unifying all of them and always indicatingwith one or two decimal places (including percentages).
Reply: Thank you for the comment. We have made the respective ammendments.

Comment: The comments that the authors make in the results section about them should beeliminated, such as in line 156 "As expected...", as well as the paragraph thatbegins in line 169. In the results section only They must provide the data and the authors must not make any comments on them; Comments should be reserved fordiscussion.
Reply: Thank you for the comment. We have made the respective ammendments.

Comment: In tables 1 and 2, the percentages of the categorical variables should be entered.
Reply: Thank you for the comment. We have made the respective ammendments.

Comment: No relationship test is done to show causality or influence, such as providing OR data.
Reply: Thank you for the comment. We did not perform such analysis.

Comment: Logically, patients who have been on CPAP for 6 months are more likely to continue with it. I don't understand the phrase: "CPAP compliance at 1 year is higher in patients already fully compliant at 6 months." Higher than what? Patients who were not compliant at 6 months were already off cpap.
Reply: Thank you for the comment. We have changed our initial overwhelming phrase.

Reviewer 3 Report

Comments and Suggestions for Authors

Interesting subject, i am not sure if i has red an article studying use of CPAP during covid 19.

nevertheless, study needs few major clarifications:

1-In CPAP, is there any algorithms able to secure artefacts due to extrinsec input/stimuli?

2-it has been recently discover that the influence of socioeconomic status (through indicators like education, income, neighborhood, etc…) can affects affects directly quantitative results measured by objective measurements like polysomnography and actigraphy. Majority of this study were performed with CPAP, so pretty similar to what issue authors have faced or may faced. In addition, Covid 19 itself may affects a lot prevalence of breathing disorders. I suggest authors to consider following articles for discussion:

a) socioeconomic position and excessive daytime sleepiness: a systematic review of social epidemiological studies

b) Excessive sleepiness and associated symptoms in the U.S. adult population: Prevalence, correlates, and comorbidity.

Author Response

Comment: Interesting subject, i am not sure if i has red an article studying use of CPAP duringcovid 19.
nevertheless.
Reply: Thank you for your compliment.

Comment: Study needs few major clarifications: In CPAP, is there any algorithms able to secure artefacts due to extrinsecinput/stimuli?
Reply: Monitoring using ResMed Airview is fully compliant with sleep standards and is trustworhty. Monitoring at follow-up visits is based on data from the CPAP machines. We are not aware of any other factors that could impact our data and information. So, the data used are safe and trustworhty.

Comment: it has been recently discover that the influence of socioeconomic status (throughindicators like education, income, neighborhood, etc…) can affects affects directly quantitative results measured by objective measurements like polysomnographyand actigraphy. Majority of this study were performed with CPAP, so pretty similar to what issue authors have faced or may faced. In addition, Covid 19 itself may affects a lot prevalence of breathing disorders. I suggest authors to consider following articles for discussion:
a) socioeconomic position and excessive daytime sleepiness: a systematic reviewof social epidemiological studies, b) Excessive sleepiness and associated symptoms in the U.S. adult population:Prevalence, correlates, and comorbidity.
Reply: Thank for your comment. We have made the respective ammendments.

Round 2

Reviewer 1 Report

Comments and Suggestions for Authors

The authors diligently revised the manuscript according to the suggestions of the reviewers, leading to a marked improvement in the quality of the article.

Author Response

Comment: The authors diligently revised the manuscript according to thesuggestions of the reviewers, leading to a marked improvementin the quality of the article.

Reply: We would like you for the time spent and effort made to provide us with your valuable comments and advice.

Reviewer 2 Report

Comments and Suggestions for Authors

I thank the authors for the effort they have made to follow the reviewer's recommendations.

The authors have not satisfactorily answered the methodological doubts raised. Likewise, certain errors remain uncorrected, such as commenting on the results in the results section and not in the discussion.

For all these reasons, the recommendation continues to be that the paper be rejected.

Author Response

Comment: I thank the authors for the effort they have made to follow the reviewer's recommendations.

Reply: Thank you for your comment.

Comment: The authors have not satisfactorily answered the methodological doubts raised. Likewise, certain errors remain uncorrected, such as commenting on the results in the results section and not in the discussion.

Reply: We can understand your previous concerns on methodology, and we have provided answers and made changes in the previous revision.

Comment: For all these reasons, the recommendation continues to be that the paper be rejected.

Reply: We can't comment on your decision.